



# Simulated Ka- and Ku-band radar altimeter scattering horizon on snow-covered Arctic sea ice

Rasmus T. Tonboe[1], Vishnu Nandan[2,3], John Yackel[3], Stefan Kern[4], Leif Toudal Pedersen[5] , Julienne Stroeve[2]

[1]Danish Meteorological Institute, Copenhagen, Denmark
[2]Centre for Earth Observation Science (CEOS), University of Manitoba, Canada
[3]Cryosphere Climate Research Group, University of Calgary, Canada
[4]University of Hamburg, Germany
[5]Technical University of Denmark, Denmark

*Correspondence to*: Rasmus T. Tonbe (rtt@dmi.dk)

**Abstract.** Owing to differing and complex snow geophysical properties, radar waves of different wavelengths undergo variable penetration through snow-covered sea ice. However, the mechanisms influencing radar altimeter backscatter from snow-covered sea ice, especially at Ka- and Ku-band frequencies, and its impact on the Ka- and Ku-band radar scattering horizon or the 'track point' (i.e. the scattering layer depth detected by the radar re-tracker), are not well understood. In this study, we evaluate the Ka- and Ku-band radar scattering horizon with respect to radar penetration and ice floe buoyancy using a first-order scattering model and Archimedes' principle. The scattering model is forced with snow depth data from the European Space Agency (ESA) climate change initiative (CCI) round robin data package, NASA's Operation Ice Bridge (OIB) data and climatology, and detailed snow geophysical property profiles from the Canadian Arctic. Our simulations demonstrate that the Ka- and Ku-band track point difference is a function of snow depth, however, the simulated track point difference is much smaller than what is reported in the literature from the CryoSat-2 Ku-band and SARAL/AltiKa Ka-band satellite radar altimeter observations. We argue that this discrepancy in the Ka- and Ku-band track point differences are sensitive to ice type and snow depth and its associated geophysical properties. Snow salinity is first increasing the Ka- and Ku-band track-point difference when the snow is thin and then decreasing the difference when the snow is thick (>10 cm). A relationship between the Ku-band radar scattering horizon and snow depth is found. This relationship has implications for 1) the use of snow climatology in the conversion of radar freeboard into sea ice thickness and 2) the impact of variability in measured snow depth on the derived ice thickness. For both 1 and 2, the impact of using a snow climatology versus the actual snow depth is relatively small on the measured freeboard, by only raising the measured freeboard by 0.03 times the climatological snow depth plus 0.03 times the real snow depth. This study serves to enhance our understanding of microwave interactions towards improved accuracy of snow depth and sea ice thickness retrievals from combining currently operational and upcoming Ka- and Ku-band dual-frequency radar altimeter missions, such as ESA's Copernicus High Priority Candidate Mission CRISTAL.



## 1 Introduction

Since 2010, basin-scale Arctic sea-ice thickness ($H_I$) has been estimated monthly during the winter season using European Space Agency's CryoSat-2 Ku-band frequency radar altimeter data. (e.g. AWI https://www.meereisportal.de, UCL http://www.cpom.ucl.ac.uk/, and NSIDC https://nsidc.org/data/RDEFT4) and from Ka-band SARAL/AltiKa radar altimeter data (e.g. Maheshwari et al., 2015). Neither CryoSat-2 nor AltiKa directly measure $H_I$. Instead, they provide a measure of the sea ice freeboard ($F_I$) — the height of the sea ice floe from the local sea level, either measured in leads or cracks located

adjacent to the floe. To convert $F_I$ to $H_I$, hydrostatic equilibrium is assumed (Laxon et al., 2003). This assumption requires geophysical property information on the overlying snow pack as well as the underlying sea ice, which can affect the accuracy of the radar scattering horizon. These geophysical parameters include snow depth, snow density, temperature, salinity, snow grain size, snow surface/sea ice interface roughness, sea ice density and sea water density (Landy et al., 2020; Nandan et al., 2020; Landy et al., 2019; Tonboe et al., 2010; Nandan et al., 2017; Alexandrov et al., 2010; Ricker et al.,

2014). The radar scattering horizon or track point is conceptualized as the scattering surface depth detected by the radar re-tracker algorithm and the floe buoyancy; and in turn impacts the accuracy of $F_I$ and $H_I$ estimates (Ricker et al., 2014). The scattering horizon or track point represents the return radar echo waveform measured by a radar altimeter, which is then statistically analyzed in the re-tracker algorithm to extract information on the scattering surface depth between the air/snow interface and a physical interface either within the snow pack volume, at the snow/sea ice interface or within the sea ice

volume (e.g. Ricker et al., 2014). However, the detected horizon may not coincide with a physical interface. That is why we prefer to call it the 'track point'. The re-tracker algorithm can be tuned so that the radar scattering horizon coincides with the snow/sea ice interface. However, satellite radar backscatter interactions are non-linear, and the total backscatter is dominated by a relatively small areal fraction of plane facets on the surface (Fetterer et al., 1992; Ulander and Carlström, 1991). Also, thinner ice types exhibit higher backscatter than thicker sea ice because of differences in surface roughness, leading to

preferential sampling of the thinner ice types (Tonboe et al., 2010; Aldenhoff et al., 2019). In other words, the radar scattering is dominated by an area which is only a fraction of the total surface and bulk snow and ice properties which are relevant for the buoyancy of the floe and may not be representative for the scattering parts of the floe. This has implications for how one might acquire snow and ice samples in the field and how snow depth and ice thickness and density are used in the processing of radar altimeter data for deriving $H_I$. When deriving $H_I$ snow depth, snow, ice and water density and radar

penetration are accounted for in the processing of the sea ice thickness products from CryoSat-2 (e.g. Hendricks et al., 2016). During the $F_I$-to-$H_I$ conversion and if the snow depth is known there are two corrections involving snow: 1) there is a radar penetration correction that will compensate for this sensitivity and locate the scattering horizon at the snow/sea ice interface (Kwok et al., 2011; Ricker et al., 2014; Mallett et al., 2020), and 2) there is a correction to the ice floe buoyancy as a function of snow depth.




Several studies suggest that it may be possible to derive snow depth directly using a dual-frequency approach by combining Ka- and Ku-band radar altimetry (e.g. Lawrence et al. 2018; Guerreiro et al. 2016). The underlying principal behind this technique is that the assumption of predominant Ka-band scattering originates at the air/snow interface, while for Ku-band, the dominant scattering originates at the snow/sea ice interface (Beaven et al., 1995; Lawrence et al., 2018; Laxon et al.,

2013; Kurtz et al., 2014). Armitage and Ridout (2015) compared the effective scattering surface of the Ka-band altimeter AltiKa and Ku-band CryoSat-2 to the snow depth and snow surface measurements from NASA's Operation Ice Bridge (OIB) campaigns. They found that the AltiKa dominant scattering horizon is 0.54 times the snow depth above the ice surface using the OIB Quick Look snow depth product. They also found that the CryoSat-2 radar scattering horizon was deeper into the snow volume, well below the Ka-band scattering horizon, but still above the snow/sea ice interface and that the depth of

this horizon was dependent on sea ice type. Observed AltiKa and CryoSat-2 mean freeboard differences were found to be ~ 4 to 7 cm from October to March (Fig. 2 in Armitage & Ridout, 2015). Lawrence et al. (2018) indicated that some of these differences between AltiKa and CryoSat-2 could be attributed to the different re-tracker algorithms used in the processing scheme of the two datasets.

While Guerreiro et al. (2016) found that Ka-band radar scattering primarily originates from the air/snow interface, based on simple modeling assumptions, Maheshwari et al. (2015) assumed the effective Ka-band scattering interface was coincident with the snow/sea-ice interface in their derivation of sea ice freeboard using AltiKa. Seasonally evolving snow covers with internal density layering (e.g. compacted wind slabs), ice lenses, melt-refreeze layers, brine-wetting (only on first-year sea ice) and large spatial diversity, adds to the geophysical complexity and manifests vertical shifting of the Ku-band radar

scattering horizon by several or more centimeters above the snow/sea ice interface (Nandan et al., 2017; Tonboe et al., 2006b). This significantly impacts the accuracy of $F_I$ and $H_I$ retrievals from radar altimetry, both in the Arctic and in the Antarctic (Nandan et al., 2020; Kwok and Kacimi, 2018; Ricker et al., 2014; Ricker et al., 2015; Kwok et al., 2014; Hendricks et al., 2010). This ambiguity and inconsistency in assumptions and previous study results suggests detailed investigation into the location of the Ka- and Ku-band radar scattering horizons for snow-covered sea ice is warranted.


In this study, we simulate the combined effect of snow depth and density on the Ka- and Ku-band radar scattering horizon and on the sea ice floe buoyancy. To achieve our research objective, we use a radar scattering model, together with a reference snow depth and density dataset from the European Space Agency (ESA) climate change initiative (CCI) round robin data package programme, to describe any potential variability in Ka- and Ku-band radar scattering horizon in snow-

covered Arctic sea ice. For the scattering model, we use simple snow and sea ice geophysical property profiles to elucidate the Ka- and Ku-band radar scattering processes at the primary interfaces, i.e. the air/snow and snow/ice interfaces, so that we can assess the direct effect of snow depth from the Ka- and Ku-band track point difference without the influence of any other parameters which may be related to snow depth. In addition, we include five simulations from detailed snow geophysical property profiles sampled from select locations in the Canadian Arctic to assess the effect of snow density layering, snow





grain size variability and salinity variability, observed in naturally occurring snow covers on first-year sea ice. Together with the radar scattering model, we apply Archimedes principle to compute the effect of snow on the buoyancy for a snow-covered ice floe in hydrostatic equilibrium. For the simplest case, we assume a uniform snow layer on top of a uniform ice layer where $H_I$ is given as a function of $F_i$ (the sea ice freeboard is synonymous with the snow/ice interface) and snow depth ($H_S$):


$$H_I = F_I \left( \frac{\rho_{water}}{\rho_{water} - \rho_{ice}} \right) + H_S \left( \frac{\rho_{snow}}{\rho_{water} - \rho_{ice}} \right) \tag{1}$$

where $\rho_{water}$, $\rho_{ice}$ and $\rho_{snow}$ are the densities of seawater, ice and snow, respectively. Typical values from the literature for the densities of seawater, multi-year ice (MYI), first-year ice (FYI) and snow are 1024 kg m$^{-3}$, 882 kg m$^{-3}$, 917 kg m$^{-3}$ and 300 kg m$^{-3}$, respectively and these values are also used in the processing of satellite altimeter data (Laxon et al., 2013; Ricker

et al., 2014; Alexandrov et al., 2010). During the $F_I$-to-$H_I$ conversion using (1), the different assumptions regarding FYI and MYI densities translates into a 25 % $H_I$ difference between the two ice types. However, in our simulations the ice density is fixed at the density of FYI (917 kg m$^{-3}$). The snow density is varied together with the snow depth. While ice density affects ice floe buoyancy, it is not expected to influence the scattering surface depth.

**2 The ESA CCI round robin data package and snow profiles on sea ice**

The ESA Climate Change Initiative (CCI) round robin data package (RRDP) (Laxon et al., 2016) is a collection of spatially collocated and resampled Operation Ice Bridge (OIB) data (OIB version IDCSI4, 2009-2013, from NSIDC), coincident with CryoSat-2 and ENVISAT radar freeboard data, and Warren et al. (1999) snow climatology. This means that the OIB snow depth data from March and April spring campaigns are paired with the snow bulk densities from the March and April Warren et al. (1999) (W99) climatology. Since OIB flights preferentially sampled MYI in the Lincoln Sea and both FYI and

MYI types in the Beaufort Sea during March and April from 2009 to 2013, the RRDP data are representative of both dominant ice types in the Arctic. OIB snow depth and W99 snow density distributions from the RRDP data collection for both MYI and FYI are shown in Figures 1 and 2.





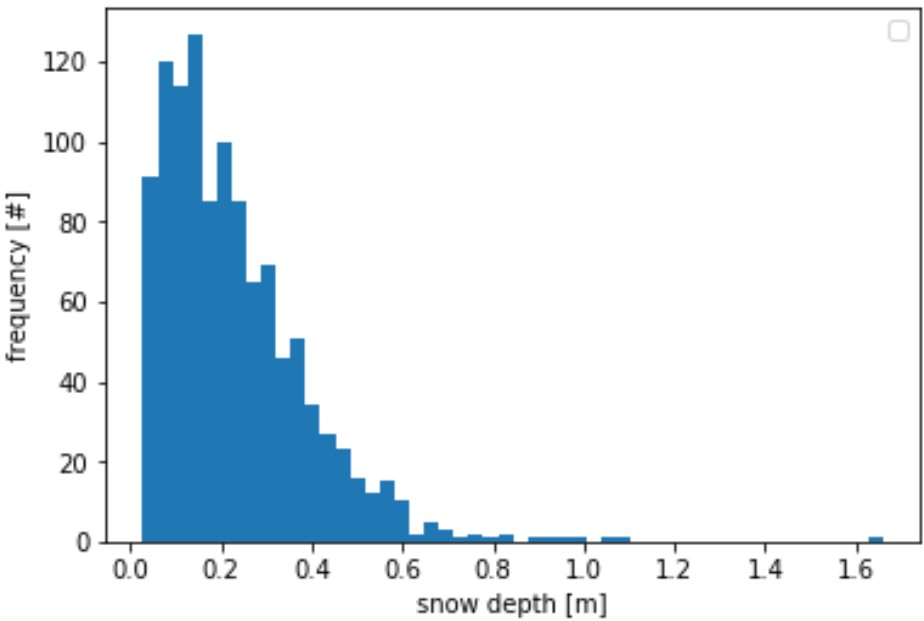

**Figure 1. March and April 2009 to 2013 snow depth data from OIB data in the RRDP dataset (N=1114) used as input to the**
**scattering model. Mean snow depth is 0.23 m and the standard deviation is 0.16 m. The minimum snow depth is 0.027 m.**

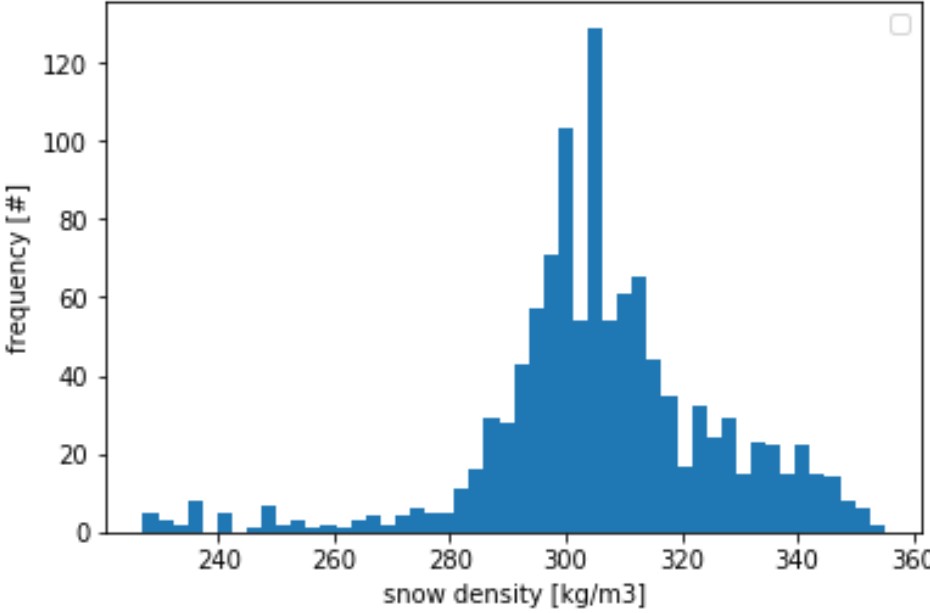

**Figure 2. Snow density distribution (N=1114) in the RRDP dataset (from W99 climatology) used as input to the scattering model.**
**Mean snow density corresponding to March and April is 306 kg m$^{-3}$ and the standard deviation is 20 kg m$^{-3}$.**






### 2.1 Snow pit data

Vertical heterogeneity of snow properties can play a significant role in accurately determining the location of Ka- and Ku-band scattering horizons (Ricker et al., 2014). Since the RRDP lacks information on this vertical heterogeneity, we

performed additional simulations using in-situ measured snow geophysical property profiles (snow salinity, temperature and density measurements sampled at 2 cm vertical intervals) acquired from five disparate snow covers acquired from the Canadian Arctic that ranged in mean thickness from 5 to 31 cm. These profiles were sampled from relatively smooth, land-fast FYI in May 2012 (late-winter season), located near Resolute Bay, Nunavut (74.70° N, 95.63° W). The in-situ drill hole measured ice thicknesses varied between 1.3 m and 1.7 m. We do not have coincident in-situ measured microscale surface

roughness estimates from these locations, but synthetic aperture radar imagery acquired from RADARSAT-2 suggests that each of the 5 samples were acquired from level and very smooth FYI. Here, we assume level sea ice and snow cover, with a flat-patch-area of 1 % and as a result surface roughness is assumed to not influence the scattering horizon variability in our model simulations. The concept of the flat-patch-area is described in the section describing the radar altimeter scattering model below.

Snow temperature was measured in situ using a Digi-Sense RTD thermometer probe (resolution of 0.1° C and accuracy ±0.2° C). Snow density was sampled using a 66.35 cm$^3$ density cutter and weighed on a Gram Precision GX-230 scale (accuracy of ±0.01 g). Snow salinity was measured in melted temperature stabilized samples using a WTW Cond 330i conductivity meter (accuracy of ±0.5 %). The samples were extracted from the snow pack with the density cutter to ensure a comparable sample volume in every sample. Snow grain radius was measured and categorized from disaggregated grain

photographs on a 2 mm grid crystal plate following Langlois et al. (2010). The snow grain size and density is used to compute the snow correlation length in Eq. 3 below.  The 5 profiles where the temperature, snow salinity and the correlation length are shown in Figure 3 are as follows:

Profile 1) 5 cm cold (snow surface temperature = -12.7° C), highly saline (7.5-14.5 ppt) snow pack with a relatively uniform

density distribution (320-360 kg m$^{-3}$).

Profile 2) 11 cm cold (snow surface temperature = -7.4° C) snow pack, saline at the bottom (14.1 ppt), and nearly non-saline at the top (0.1 ppt). Snow densities in the upper layers are 350 kg m$^{-3}$ and 250 kg m$^{-3}$ towards the basal layers. The basal layer snow densities and grain sizes indicate the presence of depth hoar.


Profile 3) 15 cm cold (snow surface temperature -12.7° C) and saline (top to bottom 3 - 13.3 ppt) snow pack. The top 11 cm layers have high densities from 400 - 430 kg m$^{-3}$ and the lowest 4 cm have densities from 220 to 250 kg m$^{-3}$. Similar to Profile 1, the bottom layer densities and the grain sizes also indicate the presence of depth hoar crystals.



Profile 4) 23 cm cold (snow surface temperature -13.5° C) non-saline snow pack. The topmost 19 cm have low densities (174 - 267 kg m$^{-3}$) while the bottommost 4 cm has higher densities (330 - 350 kg m$^{-3}$). There are layers with coarse grained snow.

Profile 5) 31 cm, almost isothermal (-2.8° C to -4.1° C), highly layered snow pack. Density varies between 226 and 877 kg

m$^{-3}$ (icy layers). The bottommost salinity contains up to 5.8 ppt but the top 20 cm of the snow profile is non-saline.

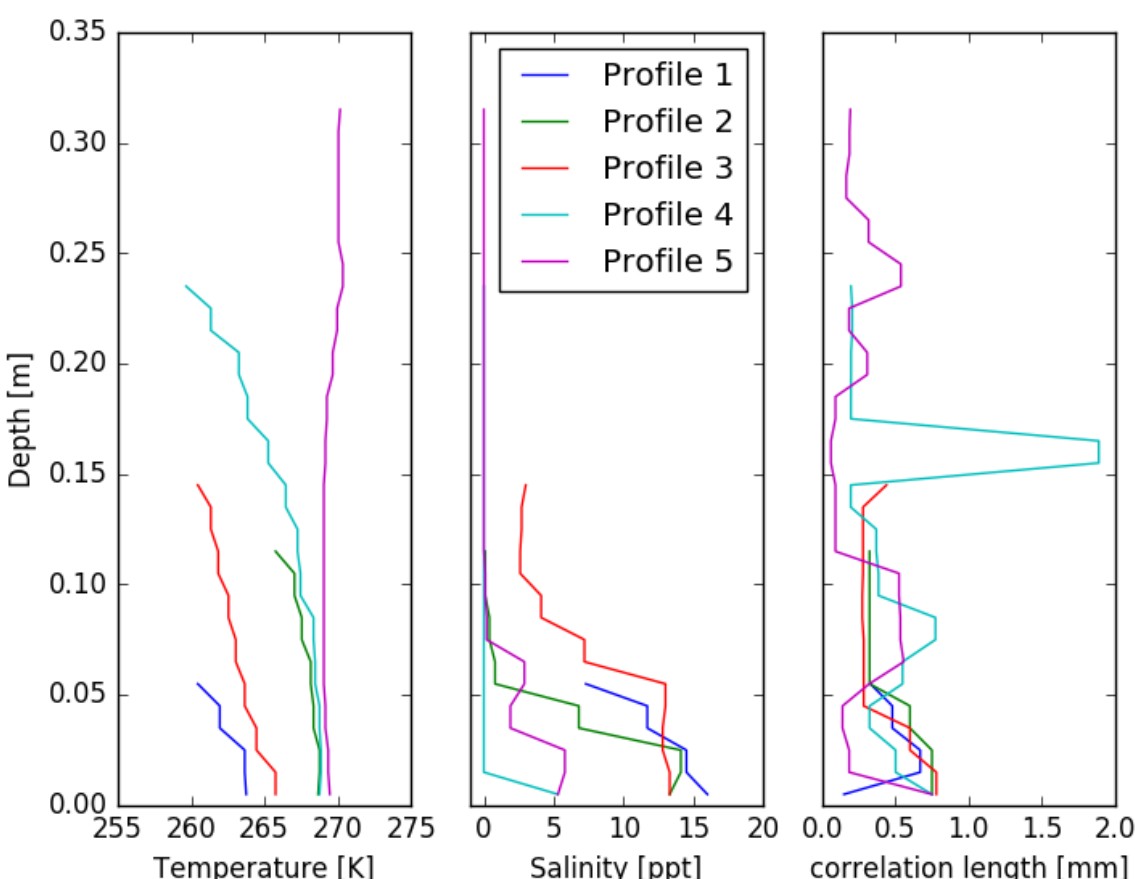

**Figure 3. Snow temperature, salinity and snow grain correlation length of the 5 depth profiles acquired from FYI in the Canadian Arctic.**


Simulations using these snow profiles (with 2 m saline FYI beneath) are compared with simulations using the uniform snow pack to assess the impact of snow density layering, snow grain size and salinity variability in naturally-occurring snow packs



on the Ka- and Ku-band scattering horizon. Our goal is to separate the direct effect of snow depth in the uniform vertical profile of geophysical properties on the Ka- and Ku-band scattering horizon, and compare them with the derived scattering

horizons influenced by the effects of layered snow packs.

## 3 Radar altimeter scattering model and re-tracker description

The radar scattering model utilized in this study is a multi-layer, one-dimensional radiative transfer model where surface scattering is computed at horizontal interfaces (snow surface, interfaces within the snow pack and ice surface), as described in Tonboe et al. (2006a; 2010; 2017), and conceptually comparable to models developed by others (e.g. Landy et al., 2019).

The multi-layer model concept is different from single layer scattering models developed for ice sheet backscatter (e.g., Ridley and Partington, 1988) since surface scattering dominates in sea ice (Ulander et al., 1991; Fetterer et al., 1992). The model — flowchart from input of the physical snow and ice profiles to computing the track point — is illustrated in Figure 4.

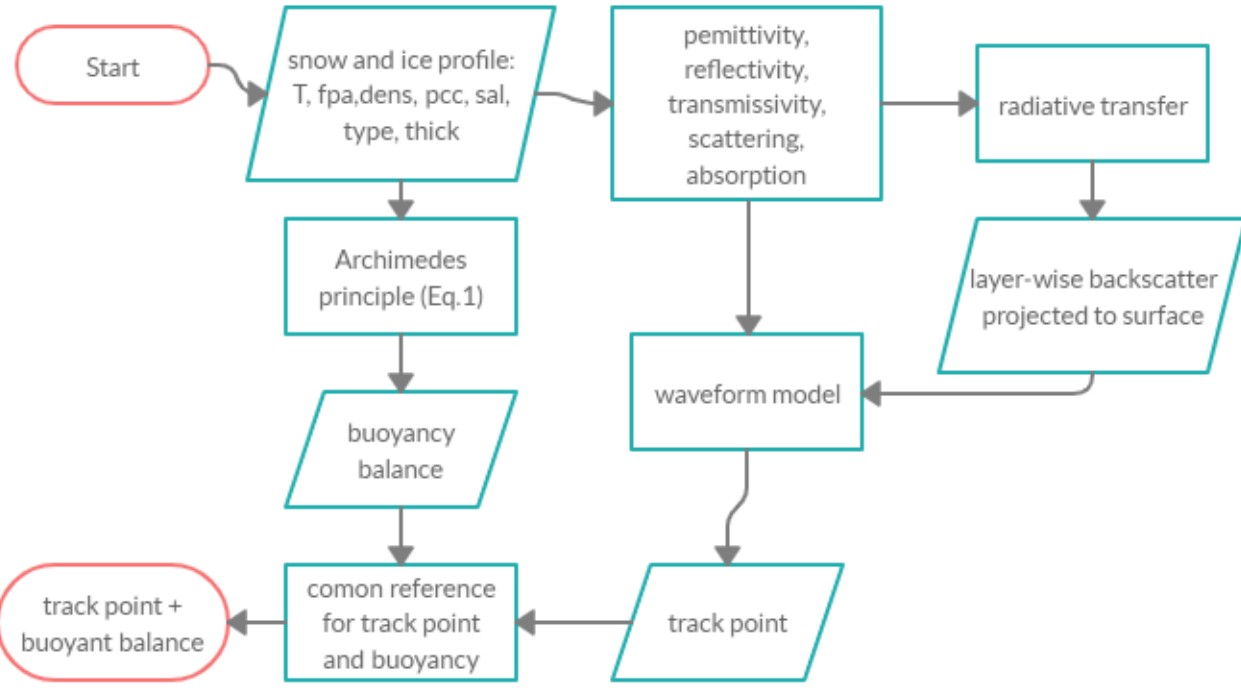


**Figure 4. Computational steps in the scattering model to reach the radar altimeter track point. The model is described in detail in Tonboe et al., 2006a and 2010. The snow and ice profile has temperature (T), flat-patch-area (fpa), correlation length (pcc), salinity (sal), snow or ice type (type), and thickness (thick) for each layer.**





The scattering model uses layer-wise information on snow/sea ice stratigraphy (layer thickness in meters), temperature (K), snow salinity (ppt), snow density (kg m$^{-3}$), correlation length (a measure of the snow grain size or the size of inclusions, e.g., brine or air, in the sea ice) (mm), interface roughness (fraction of total area), and derived brine volume from snow salinity and temperature. The model uses a radiative transfer approach to compute the total backscatter, $\sigma_{total}$ (Tonboe et al., 2010).

$$\sigma_{total} = (\sigma_i^{surf} + T_i^2 \sigma_i^{vol}) \prod_{i=1}^{n} \frac{1}{L_{i-1}^2} T_{i-1}^2 \qquad (2)$$

where $\sigma_i^{surf}$ is the interface surface scattering for layer $i$, $T_i$ is the interface transmissivity, $\sigma_i^{vol}$ is the layer volume scattering, $L$ is the layer loss (scattering and absorption). Radar propagation speed, attenuation and scattering are computed for each layer. We use a geometric description of the footprint area in each layer as a function of time for a pulse limited

altimeter and the time dependent area is multiplied with the time dependent backscatter resulting in the waveform (Tonboe et al., 2010). The track point is found at half of the maximum waveform power point in time (Tonboe et al., 2010). While different track point thresholds will shift the scattering horizon vertically (Ricker et al., 2014), the location of the scattering horizon does not change the modeled sensitivity to snow depth (Tonboe, 2017).

Since the total backscatter is dominated by surface/interface scattering, the surface scattering model used in this study assumes that the backscatter return signal is dominated by specular reflection processes from relatively small plane areas (flat-patches), normal to the near-nadir radar signal within the footprint, described in Fetterer et al. (1992; eq. 18) and Ulander and Carlström (1991). This assumption is believed to be "more realistic" than other sea ice surface scattering models because the satellite nadir/near-nadir radar backscatter is dominated by reflections from smooth patches on the

surface (Fetterer et al., 1992). The fundamental assumption for all surface scattering models is that the backscatter is a function of the reflection coefficient, interface roughness and slope i.e. at nadir, when the interface is smooth the backscatter is high, and when the surface is rough then the backscatter received by the radar is smaller.

The permittivity of the snow and ice is computed using the two-phase mixing formulas described in Mätzler (1998). The

permittivity of dry snow is primarily a function of snow density and the permittivity of sea ice and saline snow depends on salinity and temperature, i.e. brine volume and snow or ice density, (Frankenstein and Garner, 1967; Drinkwater and Crocker, 1988). The permittivity of both materials is computed using the mixing formulae for rounded spheres as inclusions in a background matrix of air or ice (Mätzler, 1998) and the equations for brine volume and permittivity in Ulaby et al. (1986). When the snow is saline, we use a formulation for wet snow in Ulaby et al. (1986) and an estimation of the brine

volume as a function of salinity, density and temperature (Ulaby et al., 1986). This is feasible since the permittivity of fresh water and brine is the same for radar frequencies larger than about 10 GHz including both Ka- and Ku-band (Ulaby et al., 1986). The predictions of different snow and ice permittivity models vary as a function of brine pocket, air bubble or snow




particle inclusion shape and permittivity (Ulaby et al., 1986). We believe that the choice of model will have an impact on the absolute magnitude of the model estimates, however, only a smaller impact on the relative variability of the model predictions. Volume scattering from snow grains or inclusions in the ice is computed for each layer using the "Improved Born Approximation" for spherical inclusions (Mätzler, 1998) and is included in our radiative transfer calculation. Although the volume scattering contribution to the overall backscatter is considered insignificant, its contribution adds to the signal extinction and therefore affects the loss factor and the track point.

We convert the optical snow grain size (described in the detailed snow profiles), to snow correlation length, $p$, which is used in the model describing the scatter size, i.e. (Mätzler, 2002),

$$p = 0.5 D_0 (1 - \nu) \tag{3}$$

where $D_0$ is the optical snow grain diameter in millimeters and $\nu$ is the bulk snow density divided by the pure ice density (917 kg m$^{-3}$).

In this study, the track point is computed as a point in time located midway between the noise floor and the maximum return signal power received by the radar. Different track point thresholds change the vertical height of the scattering horizon as described in Tonboe (2017). On ice sheets and sea ice where surface scattering dominates, the half power time re-tracking threshold provides a good estimate of the mean surface elevation (Davis, 1997).

## 4 Scattering model initialisation and set-up

The scattering model uses a multi-layer snow and sea ice profile as input. The simplest case consists of one uniform snow layer on top of an overlying uniform ice layer, with the parameters listed in Table 1 used as input to the model. While salinity, temperature and interface roughness are fixed, snow depth and density vary as given in the RRDP data set. We set the snow grain correlation length (a measure of scatter size) to 0.1 mm following Tonboe et al. (2010) and Tonboe (2017). For snow/ice salinity and temperature, we assume no salinity in the snow pack and an isothermal temperature of 263.15 K. Sea ice salinity and temperature is set at 3.0 ppt and a temperature of 269.15 K, respectively. These values represent non-melting conditions. The surface roughness is quantified as the flat-patch area which is the areal fraction of flat-patches contributing coherently to the total backscatter (Fetterer et al., 1992; Ulander and Carlström, 1991). Here, the flat-patch area is set to 0.01 for both the snow and the ice surface (Tonboe et al., 2010). With this information, the model produces the backscatter coefficient, waveform, and track points at half of the maximum power. The model then simulates the Ka- and Ku-band radar track point variability of homogeneous un-layered snow packs during winter as a function of snow depth and density only. Since both the track point and the floe buoyancy are affected by snow depth and density, the scattering model




is used together with "Archimedes' principle" to compute the sensitivity of both simultaneously. The fixed value of surface roughness used in these simulations at the snow surface and at the snow/ice interface will affect the height of the scattering

surface for both Ka- and Ku band, while the sea ice density will primarily affect the floe buoyancy's impact on the track point (Tonboe et al., 2010).

The scattering model is first initiated with uniform snow and sea ice properties and then for each subsequent simulation, the snow density and snow depth in Table 1 are exchanged with OIB snow depth and the Warren et al. (1999) snow density from

the RRDP data set in order to investigate the sensitivity to observed snow depth and density variability. Then we additionally use the 5 in-situ profiles to study the effect of snow density layering, snow grain size variability and snow salinity variability in natural snow packs on the track point.

**Table 1. Initial run input to the scattering model. T is the layer temperature, Roughness is quantified as the flat-patch area which is the fraction of specular facets compared to the total area (F), Density is the layer density, Depth is the layer thickness, Correlation length is a measure of the scatter size (and distribution), Salinity is layer salinity. Variables marked in bold in the table are exchanged with values from the RRDP for each simulation. There are 1114 data points in the RRDP data set.**

| Layer number | T [K] | Roughness F [1/100] | Density [kg m$^{-3}$] | Depth [m] | Corr. length [mm] | Salinity [ppt] | Type |
|---|---|---|---|---|---|---|---|
| 1 | 263.15 | 0.01 | **300** | **0.2** | 0.1 | 0.0 | snow |
| 2 | 268.15 | 0.01 | 917 | 2.0 | 0.2 | 3.0 | sea ice |

**5 Ka- and Ku-band altimetry track point difference simulation results and discussion**

The Ka- and Ku-band track point difference as a function of snow depth, density and correlation length is illustrated in Figure 5. We find that the Ka- and Ku-band track point difference is mostly insensitive to snow depth; i.e. the track point difference ranges between 0 cm and 8 cm, for coarse-grained (0.3 mm) snow depths between 0.05 and 0.65 m. The sensitivity increases with snow depth and is highest for snow > 50 cm in thickness. The sensitivity to snow depth decreases for smaller snow correlation lengths, such that the track point difference is about half (0 cm to 4 cm) for a snow correlation

length of 0.1 mm (red), compared to 0.3 mm (blue), keeping all other parameters unchanged (see Table 1). We are not varying the surface roughness in our experiments and to investigate the impact of the air/snow and snow/ice interface roughness on the Ka- and Ku-band track point difference we would require a different surface scattering model and data for the interface roughness. This remains outside the scope of this study.



Figure 6 illustrates the effect of the 5 snow profiles on FYI in the Canadian Arctic. Of interest is the presence of saline snow covers on FYI, which has been long recognized for its effect on microwave/radar signal propagation (Geldsetzer et al., 2007; Yackel and Barber, 2007; Nandan et al., 2020; Nandan et al., 2017; Kwok and Kacimi, 2018; Barber and Nghiem, 1999; Barber et al., 1998 and references therein). With changes in snow temperature, salinity and density in the snow layers, snow brine volume is modified towards the snow basal layers and at the snow/sea ice interface, masking the propagation of radar

waves from reaching the snow/sea ice interface (Barber and Nghiem, 1999; Nandan et al., 2017). This results in an upward shift of the track point. In our study, the simulations were rerun at 1% snow brine volume (1 % brine volume is equivalent to a bulk snow salinity of about 2 ppt at -10° C bulk snow temperature), after which, the Ka- and Ku-band track point differences were acquired. For snow covers < 10 cm, the saline snow at first increases the sensitivity of the Ka- and Ku-band track point difference to snow depth, but for snow depths > 10 cm, the signal loss in the snow cover caused by the brine

results in identical track points at Ka- and Ku-band. Snow extinction is the sum of scattering from snow grains and attenuation from brine when the snow is saline. While attenuation in the snow is comparable at Ka- and Ku-band, scattering is different, and it is the scattering contribution to the extinction which is creating the Ka- and Ku-band track point difference. Deeper snow (more scatters) and/or larger snow grains (scatters) gives more scattering and a larger Ku- and Ku-band difference by increasing extinction and the relative importance of the snow/ice interface scattering. When the depth of

saline snow is increased then the Ka- and Ku-band track point difference initially increases compared to non-saline snow. This is because the attenuation is controlling the relative importance of the snow ice interface scattering compared to the snow surface scattering which is invariant in these experiments, and again it is the scattering from the snow grains which is producing the Ka- and Ku-band track point difference. When the snow depth is > 10 cm then the relative importance of the snow/ice interface scattering is minimal and when the saline snow is ~ 40 cm deep then snow surface scattering totally

dominates and there is no Ka- and Ku-band track point difference because both radar wavelengths are scattered at the snow surface. The average sensitivity of the Ka- and Ku-band track point difference to snow depth in our simulations is small (30:1). For coarse grained snow using the mean snow depth from the RRDP dataset of 23 cm as the reference point the track point difference is 8 mm (Figure 3). This indicates that the Ka- and Ku-band track point differences observed in Lawrence et al. (2018) and in Guerreiro et al. (2016) is not only caused by the snow depth itself, but in combination with, for example,

the snow grain size and/or snow salinity or other factors that we have not investigated here. Armitage and Ridout (2015) found that the Ka- and Ku-band track point difference is a function of sea ice type as well. Snow depth is indirectly linked with sea ice type because the accumulation period is longer, generating thicker and denser snow for SYI/MYI also because SYI/MYI is rougher and it disproportionately 'traps/captures/entrains' more blowing/drifting snow than for FYI (Iacozza and Barber, 1999; Liston et al., 2019). In addition, snow cover on FYI is usually saline, especially in the bottommost 6 to 8 cm

snow layers (Drinkwater & Crocker, 1988; Barber et al., 1998; Nandan et al., 2017), and this will affect the Ka- and Ku-band track point difference.





Sensitivity of the Ka- and Ku-band track point difference to variations in snow pack properties from our 5 profiles is summarized in Table. 2. The track point difference is essentially zero when the snow is saline. However, profile 4 (non-

saline snow pack with depth of 23 cm) has a Ka- and Ku-band track point difference of 5.8 cm which is comparable to differences reported by Armitage and Ridout (2015). This illustrates that the track point difference can be higher for naturally observed snow profiles than for the uniform profile results shown in Figure 5. In addition to being non-saline, profile 4 has layers with coarse grained snow. The snow correlation length in these layers is much larger than for any of the other profiles. The scattering magnitude contributing to the radar signal extinction in the snow at Ka- and Ku-band is very

different and this is affecting radar penetration and consequently the track point difference in profile 4.

**Table 2. Summary of the Ka- and Ku-band track point difference for 5 snow profiles on FYI in the Canadian Arctic.**

|  | Profile 1 | Profile 2 | Profile 3 | Profile 4 | Profile 5 |
|---|---|---|---|---|---|
| Depth and salinity characteristics | 0.05 m saline snow pack | 0.11 m where bottom snow pack is saline | 0.15 m saline snow pack | 0.23 m non-saline snow pack with coarse grained snow | 0.31 m where bottom of the layered snow pack is saline |
| Ka-Ku track point difference [m] | 0.004 | 0.009 | 0.027 | 0.058 | 0.017 |



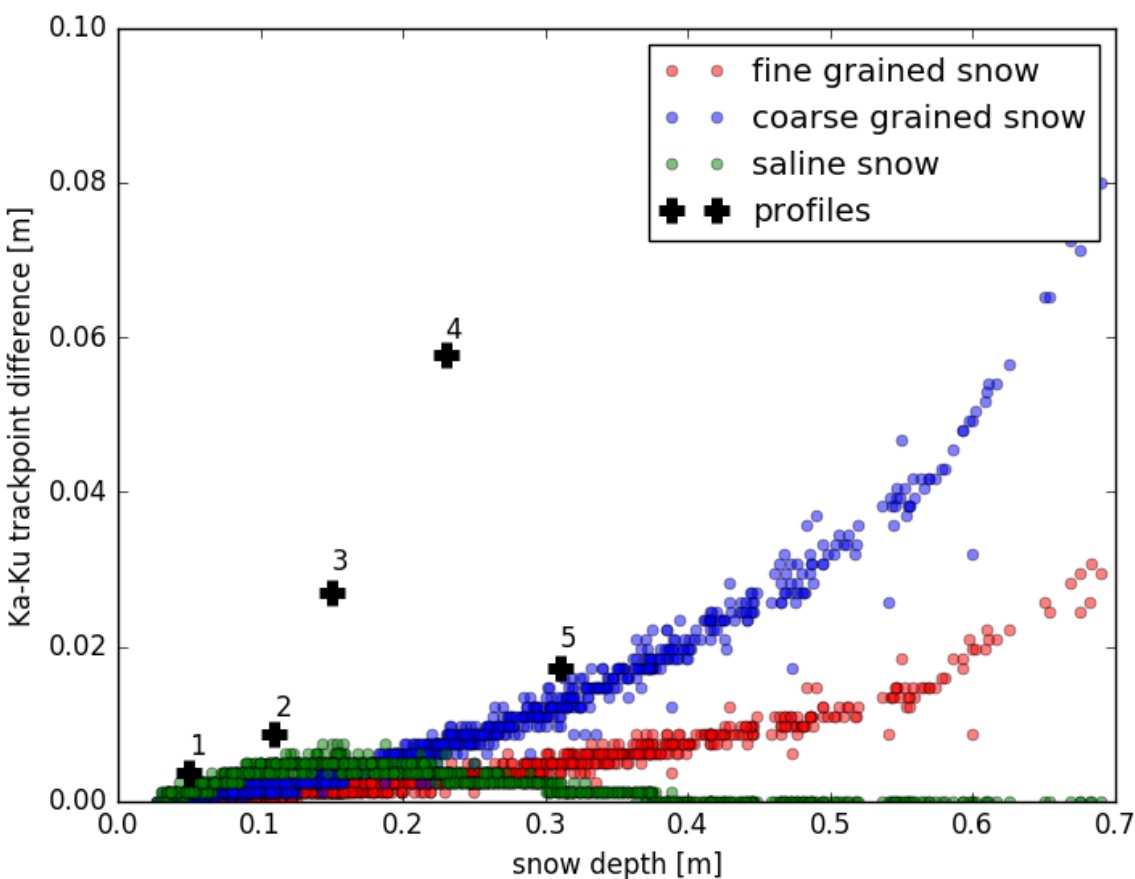

**Figure 5. The Ka- and Ku-band track point difference as a function of snow depth (and density). The red points represent the profile in Table 1, while blue points represent coarse grained snow (correlation length: 0.3 mm) and green points represents saline snow (salinity 2 ppt). The five simulated profiles in Table 2 are marked with black crosses and the numbers refer to the profile number in Table 2.**

## 6 Snow climatology for radar sea-ice freeboard to thickness conversion

It is common practice in sea ice altimetry to use the W99 snow climatology in the $F_I$ to $H_I$ conversion (Laxon et al., 2013; Kurtz and Farrell, 2011). The snow climatology is used to 1) compensate for the effect of the snow cover on the ice floe buoyancy, and 2) to compute radar propagation in the snow. In practice, on a location specific basis, the snow climatology only introduces an offset in the sea ice thickness estimation since the climatology does not reflect actual spatial and temporal snow depth and density variability.





Figure 6 summarizes the Ka- and Ku-band radar track points computed with the scattering model and the snow/sea ice interface computed from the buoyancy of the profile as a function of the snow depth and density using (1) uniform profile (Table1); (2) uniform profile with varying snow depth and density from the RRDP data; and (3) the snow profiles from the

Canadian Arctic. We do not show the actual ice thickness, but it is proportional to the freeboard. The effect of snow density is negligible because its variability in the RRDP dataset is small (mean snow density is 306 kg m$^{-3}$ and the standard deviation is 20 kg m$^{-3}$, see Figure 2). Moreover, the standard deviation of the snow density in the RRDP dataset is small, compared to other studies (e.g. King et al., 2020). Linear fits to each of the clusters are shown. The snow profiles from the Canadian Arctic, exhibiting larger vertical variability in snow density, are in close agreement with the fitted simulations for the

reference snow profile.

The green line in Figure 6 shows the combined effect of snow on the track point and the floe buoyancy. The slope is small (f = 0.03 x SnowDepth + 0.15) (SnowDepth is referred to by 'SD' in Figure 6 regression equations), suggesting that the combined effect of the radar track point and the floe buoyancy variability is almost independent of snow depth. Therefore,

the correction for snow on buoyancy and track point are almost equal and opposite in magnitude. This means that if actual snow depth information is available then the radar freeboard should be corrected for both the track point and buoyancy variation before computing sea ice thickness on a location specific basis. The effect of snow depth on the Ku- and Ka- track point is linear up to snow depths of ~ 50 cm (Figure 6). Therefore, even if the RRDP data is not fully representative of the Arctic, the results would still be valid for most of the Arctic because snow depth on Arctic sea ice is usually < 50 cm. The

"correction" for the track point is on average 0.35 times the snow depth (slope of red line in Figure 6) for a snow density of 306 kg m$^{-3}$ (standard deviation 20 kg m$^{-3}$). This is comparable to the correction used in CryoSat-2 operational processing (Tilling et al., 2018),

$$\delta h = 0.25 SD \qquad\qquad (4),$$

so that the freeboard correction $\delta h$, is 25 % of the snow depth $SD$. This equation is valid for a snow density of 300 kg m$^{-3}$. The buoyancy correction in our simulations is on average 0.28 times (slope of the blue line in Figure 6) the snow depth with an opposite sign (+/-) to the track point correction. This is equivalent to the buoyancy correction described in Eq. 1 for both FYI and SYI for a snow density of 300 kg m$^{-3}$. The snow profiles from the Canadian Arctic, with a range of snow depths (5 - 31 cm), show similar pattern as the uniform profiles for both the floe buoyancy and track point (Figure 6).


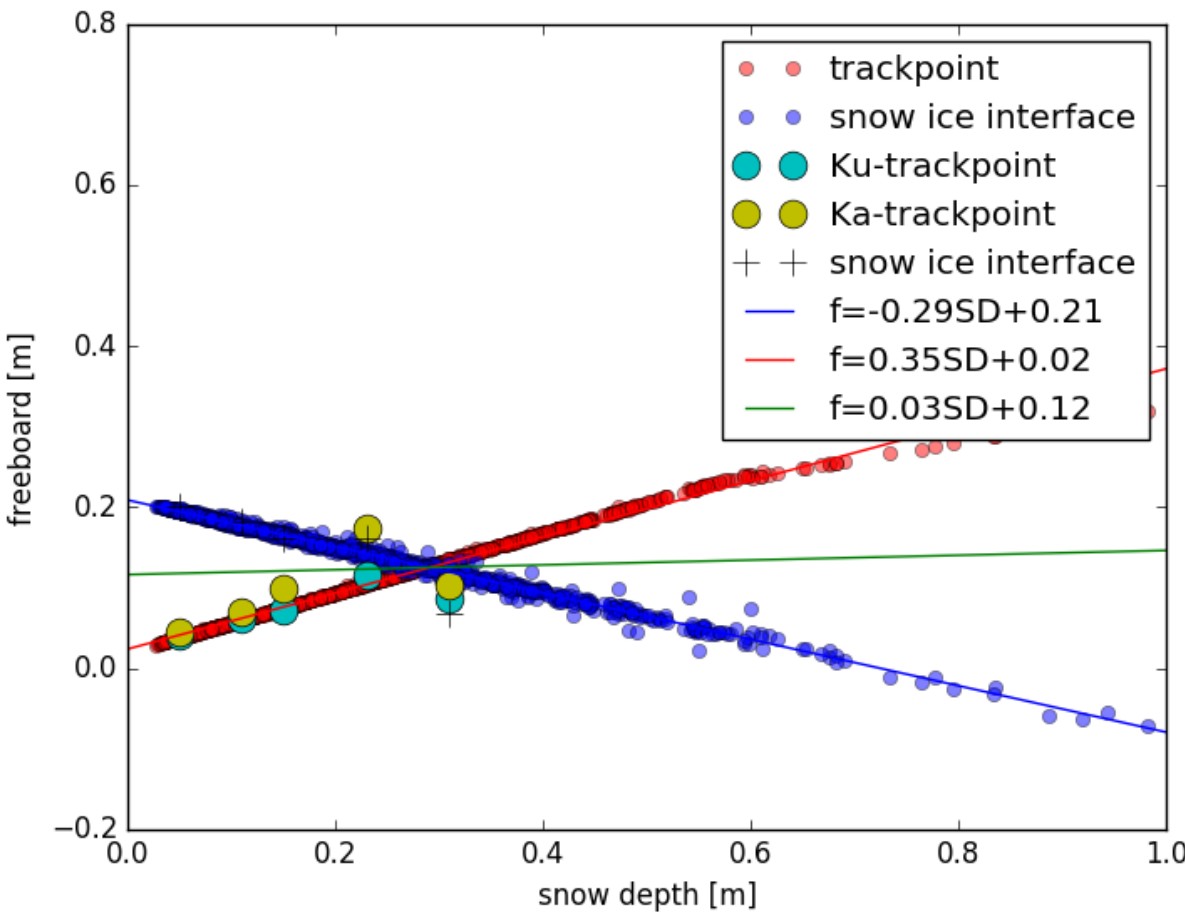

**Figure 6. Red circles is the Ku-band radar track point as a function of snow depth and density, the linear fit (red line) is the freeboard, $f = 0.35SD + 0.02$, blue circles is the snow/ice interface freeboard as a function of snow depth and density, the linear**
**fit (blue line) is $f = -0.29SD + 0.21$. The lines cross at a snow depth of 0.30 m (the physical meaning of the crossing point is related to the sea ice and sea water densities and the surface roughness and the radar frequency), the combined effect of both Ka- and Ku-band track point and buoyancy is the green line freeboard, $f = 0.03SD + 0.12$ (The figure is a reproduction of Fig. 3C in Tonboe et al. (2010) with new input data). Snow profiles from the Canadian Arctic are added, depicted with black cross marks for the snow/sea ice interface, cyan and yellow circles for the Ka- and Ku-band track points, respectively (Table 2).**


The W99 snow climatology used to convert $F_I$ to $H_I$ is seasonally and regionally varying. This means that the spatial and temporal offset that the W99 dataset introduces in the sea ice thickness derivation has regional and seasonal variability. However, this variability may not coincide with actual snow depth and density. With increasingly earlier Arctic sea ice melt onset and longer melt seasons (e.g. Stroeve and Notz, 2018), sea ice freeze-onset and snow accumulation time has also
reduced (Webster et al., 2014; Webster et al., 2018). Deviations in snow depth and density from climatology are mapped directly as systematic errors into the derived sea ice thickness changes. Climatology is used when the real snow depth is



unknown and the offset that the climatological snow depth is introducing to the freeboard measurement is 0.03 times the snow depth (the green-line slope in Fig. 6). Additionally, we need to add 0.03 times the real snow depth (if the real snow depth is unknown). This has two important implications: 1) the snow climatology results in a small impact on the derived sea

ice thickness because the radar penetration and the buoyancy correction have opposite signs (-/+), and 2) the impact of the snow on the measured freeboard is small (0.03 times the snow depth, slope of the green line in Figure 4) and this small bias is introduced by the freeboard sensitivity to snow coming from the climatology and the actual snow cover variability. The small impact of the snow on the measured freeboard is the reason why the sea ice thickness can be derived using radar altimeters even without actual snow information (current operational situation). It is also the reason why corrections using

snow climatology are relatively small compared to other errors. Other factors related to snow could influence the buoyancy and the radar scattering, e.g. snow salinity and density (e.g. Nandan et al., 2017; Nandan et al., 2020), snow grain size, roughness (e.g. Tonboe et al., 2010; Landy et al., 2020) and snow layering; and these topics warrant further research.

## 7 Conclusions

In this study we have shown that it is necessary to correct the sea ice freeboard measured by a radar altimeter for both the

snow load from actual snow depth estimates and the radar signal penetration before computing the sea-ice thickness. As a result, we advocate to avoid the use of snow climatology. We used a radar altimeter scattering model forced with snow depth and density from the European Space Agency's RRDP data set and in situ measured snow geophysical profiles obtained from land-fast FYI in the Canadian Arctic.

Our simulations demonstrate that the direct Ka- and Ku-band track point difference sensitivity is about 0.033 times the snow depth using the average snow depth of 23 cm as a reference point. This is smaller than previously reported from SARAL/AltiKa Ka-band and CryoSat-2 Ku-band track point differences of ~4 to 7 cm from October to March over the AltiKa region of coverage (e.g. Armitage and Ridout, 2015, Guerreiro et al., 2016, Lawrence et al., 2018). The simulated Ka- and Ku-band track point sensitivity is affected by snow grain size, snow salinity and vertical snow density heterogeneity,

in addition to the snow depth itself. However, the simulated Ka- and Ku-band track point differences do not explain all of the observed differences, and other factors, such as ice type (with corresponding snow salinity and snow grain size), likely affect the differences as well (Armitage and Ridout, 2015). Saline snow on FYI dampens the Ka- and Ku-band track point difference by masking the penetration of both Ka- and Ku-band radar waves from reaching the snow/sea ice interface. This result was found using both the uniform and detailed snow profiles as input to the model and supports the findings of

Armitage and Ridout (2015) who note that the Ka- and Ku-band track point difference is dependent on sea ice type. Scattering in the snow creates a Ka- and Ku-band track point difference by controlling the relative importance of the snow/ice interface scattering compared to snow surface scattering. This was shown for both the uniform profiles and the detailed snow profiles from the Canadian Arctic.

The buoyancy and Ka- and Ku-band track point corrections are nearly equal and opposite in magnitude. This implies that the measured freeboard is nearly independent of snow depth. The measured freeboard is elevated/ lowered by about 0.03 times the snow depth, with an increase/decrease in snow depth. This has two implications when deriving the sea ice thickness from the radar freeboard: 1) the snow depth climatology introduces a bias in the measured freeboard of 0.03 times the climatological snow depth plus 0.03 times the real snow depth, and 2) the impact of actual snow depth is small in the sea ice

thickness estimate and if the actual snow depth is unknown it is better not to correct than to use climatology for the correction.

A high inclination polar orbiting Ka- and Ku-band radar altimeter (CRISTAL) is being planned at ESA as one of six European Copernicus High Priority Candidate Missions for launch after 2026 (Kern et al., 2020). A primary objective of

CRISTAL is to improve upon the accuracy of snow and sea ice thickness estimates. We anticipate that our simulations will be useful in consolidating these applications and improve the measurement and mapping of snow and ice thickness from space.

**Author contribution**

RTT developed the model code and performed the simulations. RTT prepared the manuscript with contributions from all co-

authors.

**Acknowledgements**

RT was supported by the ESA sea ice CCI project. SK acknowledges support by the German Research Foundation (DFG) Excellence Initiative CliSAP under Grant EXC 177/2. VN was supported by Canada's Marine Environmental Observation, Prediction and Response Network (MEOPAR) Post-Doctoral funds. VN acknowledges support by the Natural Sciences and

Engineering Research Council of Canada (NSERC) Discovery and Research Tools and Infrastructure Grants to John Yackel, Randall Scharien and Brent Else; Polar Continental Shelf Project (PCSP) and the Northern Scientific Training Program (NSTP) for financial and logistic support. The Centre for Earth Observation Science (CEOS), University of Manitoba is acknowledged for financial and logistical support towards collection of field data.

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
