# Peer review of "Simulated Ka- and Ku-band radar altimeter height and freeboard estimation on snow-covered Arctic sea ice"

_The Cryosphere, 2020_

## Referee Comment (RC1) · Anonymous Referee #1 · 16 Sep 2020

— Summary —

In this study the authors use a multi-layer radar scattering model to simulate Ku- and Ka-band radar penetration into snow on winter Arctic sea ice. The model is forced with snow depth and density data from the ESA RRDP, and then geophysical snow parameters collected in situ. The authors conclude that the Ka- and Ku-band track point difference is a function of snow depth. While the manuscript was generally well-structured, I found that clarity was lacking in parts. I summarize this in my general comments below and also list some more specific comments. These should be addressed before publication. When completed, the manuscript will be highly relevant

to the sea ice remote sensing community, with the dual-frequency (Ku- and Ka-band) CRISTAL satellite being a high-priority Copernicus candidate mission.

— General comments —

> The presentation of results, and related discussion, is not always clear. This issue starts on P12. The paragraph was too long (a whole page), and still didn't sufficiently explain what is being shown in Figure 6. The contents of Figure 6 need to be clearly described (both in the text and caption) before the authors try to present an analysis. It may help to rearrange this paragraph and the following paragraph into three paragraphs providing 1.) a description of what is shown in Figure 6, 2.) analysis of the RRDP model runs, 3.) analysis of the profiles summarized in Table 2.

> The conclusions presented in Section 7 are not sufficiently justified. On P17 L400-401, the authors state that they advocate avoiding the use of a snow climatology. However, in the above paragraph they state that "the snow climatology results in a small impact on the derived sea ice thickness" and "The small impact of the snow on the measured freeboard is the reason why the sea ice thickness can be derived using radar altimeters even without actual snow information". This collection of statements seems contradictory. I'd like a more clearly structured argument and justification of the conclusions.

> I felt like the authors could "sell" their model a little more. There is novelty in their multi-layer approach and use of in situ profiles, so they should really highlight that!

— Specific comments —

P1 L19-20: Make it clear that OIB and climatology are used only as part of the RRDP data, and not as separate datasets

P1 L28-30: "radar freeboard", rather than "measured freeboard"? Either way, this sentence needs to be re-worded for clarity.

P2 L40-42, L45-46: It's confusing to state that geophysical properties of the ice impact

the radar scattering horizon. I know what the authors mean, but in remote sensing terminology, "scattering horizon" is commonly used to refer to a location within the snow pack. Therefore, a more accurate statement is that geophysical properties of ice affect radar height estimates, as height estimates depend on radar scattering horizon in the snow, and floe buoyancy. This is an important difference, which should be explained and then maintained throughout to avoid confusion.

P2 L50: Ricker et al., 2014 is an excellent paper. However, it is cited extensively throughout this manuscript. The authors should note that multiple other publications are also relevant, and some more so as they were published earlier.

P2 L55: This makes it sound like the MYI isn't sampled at all. Re-phrase to e.g. "height estimates dominated by FYI" or similar.

P2 L60: Hendricks et al., 2016 reference; you've listed all the products above already. Be care not to show too much of an AWI bias.

P5 Fig. 1 and Fig. 2: It'd be great to see these mapped too, to get a better representation of values over regions associated with different ice types

P6 L137: "These profiles were sampled from relatively smooth, land-fast FYI..." This information is repeated below so no need to include it here.

P6 150-151: State here why snow correlation length is important, otherwise its inclusion in Figure 3 is confusing until you read much further on

P6-7 Profiles 1-5: Include correlation length in profile descriptions

P7 L178-180: Final sentence is wordy and quite confusing

P8 L186: "...surface**/interface**..."

P9 Eqn (2): How does the model account for the different radar frequencies? This is a key principle of the paper and as far as I can tell, the information is missing. It needs to be really spelt out for those of us who are familiar with remote sensing, but not so

much with radiative transfer modelling.

P9 L214: Length scale of "smooth patches"? I assume they mean "smooth" on the order of the radar wavelength.

P10 L240: How do they calculate the noise floor?

P10 L243: Expand on what you mean by "surface scattering", i.e. from which surface (snow, ice, somewhere in-between)

P10 L243-244: I would like more justification as to why a 50% threshold was chosen. In the manuscript they mention their own 2010 paper and the Ricker et al., 2014 paper, but there are many other studies that suggest a different threshold is preferable.

P11 L276-277: Stating that the tracking point difference is "mostly insensitive" to snow depth could be misleading, when differences can reach up to 8 cm. This would have a significant impact on sea ice thickness estimates. In fact, I would just get rid of this sentence.

P11 L279: Change "smaller snow correlation lengths" to ""fine-grained (0.1 mm) snow depths" or similar, for consistency with the rest of the paragraph.

Figure 5: Belongs at the end of P11. This is a clear and interesting plot. However, it may not be the most insightful for sea ice altimetry applications. More useful would be a figure showing the fraction of penetration as a function of snow depth (and density), for Ku and Ka separately. This could be included as a second panel.

P15 L342-345: This description should be included when Figure 6 is first introduced

Figure 6: I can't make out the black crosses, and the legend covers some data. It'd also be useful to number the profiles again.

P17 L388-389: Isn't adding to an unknown an impossibility? This needs to be explained better.

— Technical comments —

P2 L36: "UCL" -> "CPOM"

P10 L255: ". . .\*\*Ka and Ku\*\* waveform. . ."

P11 L264: ". . .OIB snow depth and the Warren et al. (1999) snow density \*\*pairs\*\*. . ."

P12 L312: Define "\*\*second-year ice\*\* (SYI)"

P15 L352: "\*\*. . .Ka and Ku\*\* track point. . ."

---

## Referee Comment (RC2) · Anonymous Referee #2 · 22 Oct 2020

General comments:

The objective of this paper is to evaluate the capability of Ka/Ku bi-fréquency altimeters to measure the snow depth (SD) over sea-ice using a simulator. The authors tackle a particularly complex subject: what is the impact of the type of snow (salinity, density, temperature, grain size, etc.) on the performance of the measurements. The simulator is powered by measurements of terrain and its outputs are confronted with airborne measurements.

This type of work is indispensable to improve the quality of the measurements of the sea ice thickness (SIT) by satellite, and to prepare for the the Copernicus project of

the CRISTAL dual-frequency altimetry satellite, one of the first missions of which is to monitor the physics and dynamics of the sea ice.

As such, this work and the data used must be disseminated and made public.

Nevertheless, the results presented are in contradiction with several results already published and the arguments are not sufficiently convincing. Indeed, although this is not explicitly stated, this study seems to conclude that the Ku-frequency almost no penetrate the snow, no matter what are the snow caracteristics (see Figure 6).

In fact, most of the paper focuses on the _differential_ of snow penetration between Ka and Ku. Penetrations in the snow of each individual frequency is not analyzed. However, the conclusions are largely based on the measurement of the ice freeboard (FB) by means of the Ku frequency alone, a measure which appears only in this section 6 without being justified beforehand.

Figure 6 in the same section is therefore difficult to interpret. For example, it is not clear how the Ku nor the green differential curve have been obtained. For SD=0 we observe an ice freeboard of 0.2m and a Ku freeboard of 0m while it does not may have a problem of penetration in the absence of snow: these 2 measures should be equal.

This section 6 is far from insignificant because it leads to surprising conclusions, repeated in conclusion, including in particular the fact that the measurement of the SIT is little impacted by the method of obtaining the snow depth. This assertion is in contradiction with equation (1) of equilibrium which shows that the snow depth is involved in the process for about 30% of the measurement of the SIT (the density of snow being about 1/3 of that of water and the values of FB and SD being of the same order of magnitude).

Also the model implicitly assumes that the alitimeter is in LRM mode, while all Ku altimeters currently in flight are in SAR mode. The SAR mode has a much smaller footprint than the LRM mode. It is therefore less sensitive to surface roughness and

especially one cannot make the hypothesis of a retracking at 50% of the waveform (in SAR mode the retracker is between 85% to 95%).

This does not call into question the study presented because the comparison of Ka/Ku penetrations is a primordial subject that deserves to be studied whatever the the altimeter mode. But it is important to mention it. And with this perspective we would like to see more precisely what are the backscatter of each of these 2 individual frequencies according to the surfaces and interfaces considered (air/snow retrodiffusion, snow/ice and volume in snow).

Finally, this multilayer model seems to consider only one layer for snow, whereas we generally consider at least 2 layers for snow over sea ice, with a hard and dense superciel layer and a deepest layer of very metamorphosed grains of consequent dimensions (of the order of centimeters). This point should also be discussed.

I would therefore recommend to the authors to deepen the presentation of the measures carried out, and especially the model deployed and the conclusions that it brings on each of the frequencies, quite to reduce the part 6 on the results expected by altimetry.

Detailed comments:

P1 L27: I do not agree with the following sentence: "... the impact of using a snow climatology versus the actual snow depth is relatively small on the measured freeboard" that must be more clearly demonstrated (see general comments and other comments bellow).

P2 L45: "The radar scattering horizon or track point is conceptualized as the scattering surface depth detected by the radar re- tracker algorithm and the floe buoyancy" : this study should not depend on the buoyancy but only on the penetration. P2 L51: The following sentence is true only for the heuristic retrackers, not for the retrackers based on physical models: "The re-tracker algorithm can be tuned so that the radar scattering

horizon coincides with the snow/sea ice interface." P2 L54: What do you mean by : "leading to preferential sampling of the thinner ice types " ? P2 L61: when speaking of "penetration correction" do you include the speed propagation reduction into the snow ?

P6 L142: You say that the "surface roughness is assumed to not influence the scattering horizon variability in our model simulations" while the surface as a strong impact on the altimetric waveforms. Does that mean that the model do not reflect the altimetric behavior? Please comment.

P7 Fig 3: Please specify that depth=0.0 corresponds to the bottom, not to the surface! (if I dont mistake)

P9 L206: You say that "The track point is found at half of the maximum waveform power point in time". It is a true mean for LRM altimetry but physical retrackers show that this value varies according to the roughness and the specularity of the surface. For SAR altimetry the mean value is much higher. P9 L210: What do you mean by "the total backscatter is dominated by surface/interface scattering"? The interface is between the surfaces? Or it is another surface? Do you mean that the volum scattering is negligeable? In such a case it must be said/shown explicitely. P9 L213: In the sentence "This assumption is believed to be more realistic than other sea ice surface scattering" please specify which other sea ice surface scattering you are thinking off.

P10 L241: "the track point is computed as a point in time located midway between the noise floor and the maximum return signal power received by the radar." This is pertinent only for LRM altimetry.

P11 Table1: Only one layer for the snow. Is it realistic? Please comment.

P12 L297: Typo: "Ku- and Ku-"

P14 L338: "The snow climatology is used to 1) compensate for the effect of the snow cover on the ice floe buoyancy, and 2) to compute radar propagation in the snow." For

point 2) I suppose that you mean "compute radar slow down speed propagation in the snow" ?

P15 L345: "We do not show the actual ice thickness, but it is proportional to the freeboard." As shown by your equation (1) the SIT depends also on the snow load. So please could you precise the SIT used in the Fig 6. P15 L352: "The green line in Figure 6 shows the combined effect of snow on the track point and the floe buoyancy." Which track point? Ka? Ku? Until this section 6 only the ka-ku difference has been considered. P15 L357: "The effect of snow depth on the Ku- and Ka- track point is linear up to snow depths of $\sim$ 50 cm (Figure 6)." Fig 6 does not show the Ka measurement. P15 L360: "The correction" for the track point is on average 0.35 times the snow depth". Which track point? Ka? Ku? P15 L368: typo "SYI"

P16 Fig 6: "Red circles is the Ku-band radar track point as a function of snow depth and density" : The Ku FB has not been introduced beforhand, how do you obtain it? "The combined effect of both Ka- and Ku-band track point and buoyancy is the green line freeboard": how it is computed ? How do you get a nul Ku-FB for a ice-FB, without snow, of 20cm? Thus it is clearly not a problem of snow penetration!

P17 L389: "the snow climatology results in a small impact on the derived sea ice thickness": this sentence is clearly in contradiction with equation (1). See general comments. P17 L393: "The small impact of the snow on the measured freeboard is the reason why the sea ice thickness can be derived using radar altimeters even without actual snow information." If the first part of this sentence could be true, the second one is clearly false. Even if we can not measure precisely the FB, the SD does have nevertheless a strong impact on the resulting SIT! It is easy to demonstrate using equation (1) and various SD datasets. P17 L405: "Our simulations demonstrate that the direct Ka- and Ku-band track point difference sensitivity is about 0.033 times the snow": it is not (yet) a demonstration but still an assumption based on a model. Please mitigate.

P18 L421: "This implies that the measured freeboard is nearly independent of snow depth." Using Ka? Ku? Both? Please be more precise or mitigate. P18 L424: "the impact of actual snow depth is small in the sea ice thickness estimate": equ (1) shows that the SD may not be negligeable at all.

---

## Author Comment (AC1) · 22 Nov 2020

Tonboe et al. Anonymous Referee #1

— Summary — In this study the authors use a multi-layer radar scattering model to simulate Ku- and Ka-band radar penetration into snow on winter Arctic sea ice. The model is forced with snow depth and density data from the ESA RRDP, and then geophysical snow parameters collected in situ. The authors conclude that the Ka- and Ku-band track point difference is a function of snow depth. While the manuscript was

generally well structured, I found that clarity was lacking in parts. I summarize this in my general comments below and also list some more specific comments. These should be addressed before publication. When completed, the manuscript will be highly relevant to the sea ice remote sensing community, with the dual-frequency (Ku- and Ka-band) CRISTAL satellite being a high-priority Copernicus candidate mission.

Reply: Thank you for your review and pointing out where the MS could be improved. Responding to your comments has definitely improved the MS.

— General comments — > The presentation of results, and related discussion, is not always clear. This issue starts on P12. The paragraph was too long (a whole page), and still didn't sufficiently explain what is being shown in Figure 6. The contents of Figure 6 need to be clearly described (both in the text and caption) before the authors try to present an analysis. It may help to rearrange this paragraph and the following paragraph into three paragraphs providing 1.) a description of what is shown in Figure 6, 2.) analysis of the RRDP model runs, 3.) analysis of the profiles summarized in Table 2.

> The conclusions presented in Section 7 are not sufficiently justified. On P17 L400-401, the authors state that they advocate avoiding the use of a snow climatology. However, in the above paragraph they state that "the snow climatology results in a small impact on the derived sea ice thickness" and "The small impact of the snow on the measured freeboard is the reason why the sea ice thickness can be derived using radar altimeters even without actual snow information". This collection of statements seems contradictory. I'd like a more clearly structured argument and justification of the conclusions.

Reply: We don't believe we have presented contradicting statements. We think it is not necessary to include a bias in sea ice thickness estimation, even if it is small. The reason why this practice has been going on for so many years is because of the small bias and the difficulty to detect. We understand why reviewer 1 makes this

statement and we have added an additional clarifying sentence in the Conclusion to further describe what we are intending to say.

> I felt like the authors could "sell" their model a little more. There is novelty in their multi-layer approach and use of in situ profiles, so they should really highlight that!

Reply: Thank you for this comment. We have added some additional emphasis to this point in our Conclusion and Abstract.

— Specific comments — P1 L19-20: Make it clear that OIB and climatology are used only as part of the RRDP data, and not as separate datasets

Reply: It has been reformulated in the revised manuscript, so that it is clear that the OIB data and climatology are part of the RRDP. "(CCI) round robin data package, where NASA's Operation Ice Bridge (OIB) data and climatology are included," P1 L28-30: "radar freeboard", rather than "measured freeboard"? Either way, this sentence needs to be re-worded for clarity.

We have changed "measured" to "radar", and explained that the radar freeboard is a combination of radar scattering and buoyancy.

P2 L40-42, L45-46: It's confusing to state that geophysical properties of the ice impact the radar scattering horizon. I know what the authors mean, but in remote sensing terminology, "scattering horizon" is commonly used to refer to a location within the snow pack. Therefore, a more accurate statement is that geophysical properties of ice affect radar height estimates, as height estimates depend on radar scattering horizon in the snow, and floe buoyancy. This is an important difference, which should be explained and then maintained throughout to avoid confusion.

Reply: Thank you for this comment. Scattering is dominated by either the dielectric or surface roughness mismatch at interfaces such as the air/snow and the snow/ice interfaces. In general, scattering occurs at several vertically distributed interfaces as well. However, the track point or scattering horizon is not a real surface and that is

why we prefer to call it the 'track point' because then it relates to the waveform, that's how it is detected. In the revised manuscript, we have tried to explain this 'track-point' terminology is better and we agree that the use of the term "radar height estimate" instead of the "scattering horizon", is a more appropriate description because as you say, it includes the buoyancy as well.

P2 L50: Ricker et al., 2014 is an excellent paper. However, it is cited extensively throughout this manuscript. The authors should note that multiple other publications are also relevant, and some more so as they were published earlier.

Reply: Thanks for pointing out. We have made a conscience effort in include other key papers along these lines of inquiry. See, for e.g., the revised Introduction.

P2 L55: This makes it sound like the MYI isn't sampled at all. Re-phrase to e.g. "height estimates dominated by FYI" or similar.

Reply: Rephrased as suggested in the revised manuscript.

P2 L60: Hendricks et al., 2016 reference; you've listed all the products above already. Be care not to show too much of an AWI bias.

Reply: Understood. the intension was not to present an AWI bias but to mention the same procedures (involving the buoyancy correction) are used in other processing centers as well.

P5 Fig. 1 and Fig. 2: It'd be great to see these mapped too, to get a better representation of values over regions associated with different ice types

Reply: Ok, we have shown the distribution geographically in a separate Fig. 3.

P6 L137: "These profiles were sampled from relatively smooth, land-fast FYI: : :" This information is repeated below so no need to include it here.

Reply: Ok, we have deleted it here.

P6 150-151: State here why snow correlation length is important, otherwise its inclusion in Figure 3 is confusing until you read much further on

Reply: Ok, the correlation length, and why it is important, is described in connection with equation 3. We don't think that it fits into the data description. A description of the correlation length in each of the five profiles is included.

P6-7 Profiles 1-5: Include correlation length in profile descriptions

Reply: Ok, a description of the correlation length in the specific profile descriptions is included.

P7 L178-180: Final sentence is wordy and quite confusing

Reply: The sentence has been reformulated.

P8 L186: ": : :surface**/interface**: : :"

Reply: Thanks, we have added "/interface"

P9 Eqn (2): How does the model account for the different radar frequencies? This is a key principle of the paper and as far as I can tell, the information is missing. It needs to be really spelt out for those of us who are familiar with remote sensing, but not so much with radiative transfer modelling.

Reply: Each of the variables in eq.2 are frequency dependent. The reflection coefficient in the surface scattering model sigma_surf is a function of the permittivity and therefore frequency, and the interface transmissivity T is a function of permittivity which is a function of frequency. However, the volume scattering sigma_vol is most sensitive to frequency: frequency to the fourth power. The loss L is a function of scattering and absorption, therefore frequency. This has been clarified in the revised version.

P9 L214: Length scale of "smooth patches"? I assume they mean "smooth" on the order of the radar wavelength.

Reply: Yes, the facets are smooth at the given wavelength and large enough to return coherent backscatter. The model is described in detail in the references: Fetterer and in Ulander and Carlström.

P10 L240: How do they calculate the noise floor?

Reply: We have reformulated this sentence. There is no noise in our simulations, so the noise-floor is defined as zero backscatter.

P10 L243: Expand on what you mean by "surface scattering", i.e. from which surface (snow, ice, somewhere in-between)

Reply: Here surface scattering is interface scattering from any interface air/snow, icy layers in the snow, snow/ice interface. This has been clarified in the revised version.

P10 L243-244: I would like more justification as to why a 50% threshold was chosen. In the manuscript they mention their own 2010 paper and the Ricker et al., 2014 paper, but there are many other studies that suggest a different threshold is preferable.

Reply: We did conduct a simulation study (described in Tonboe, 2017 in the reference list) where different thresholds were tested and the choice of threshold did not change the sensitivity to snow depth. The text describing the choice of re-tracker has been elaborated. All the radar altimeter processing centers are using a correction for radar penetration in the snow (eq. 4). That means that all the re-trackers are sensitive to snow and our simulations with the 1/2 power re-tracker are simply confirming that.

P11 L276-277: Stating that the tracking point difference is "mostly insensitive" to snow depth could be misleading, when differences can reach up to 8 cm. This would have a significant impact on sea ice thickness estimates. In fact, I would just get rid of this sentence.

I sort of agreed with this reviewer here when I first reviewed the manuscript back in the summer, but I think the revised sentence reads more clearly and accurately now.

The simulations show that it is not the snow depth itself but rather the snow grain size/ scattering magnitude which is creating the Ka- and Ku-band track-point difference. Of course the scattering magnitude could be a function of snow depth, but also a lot of other things. We have reformulated this sentence so that this is clearer.

P11 L279: Change "smaller snow correlation lengths" to ""fine-grained (0.1 mm) snow depths" or similar, for consistency with the rest of the paragraph.

Ok, thanks.

Figure 5: Belongs at the end of P11. This is a clear and interesting plot. However, it may not be the most insightful for sea ice altimetry applications. More useful would be a figure showing the fraction of penetration as a function of snow depth (and density), for Ku and Ka separately. This could be included as a second panel.

Reply: We tried the reviewer's suggestion; however we were not happy with the outcome. We have included a figure showing the air/snow, snow/ice and ice/water interfaces together with the Ka- and Ku-band track points. The track points are close to the snow ice interface as expected but still sensitive to snow. This looks like a transect but it is only the first 100 points in the RRDP. This is a bit confusing, and therefore not included it in the revised MS.

[the figuer is attached] The figure is showing the air/snow, snow/ice and ice/water interfaces together with the simulated Ka- and Ku-band track points for the first 100 RRDP points.

P15 L342-345: This description should be included when Figure 6 is first introduced Figure 6: I can't make out the black crosses, and the legend covers some data. It'd also be useful to number the profiles again.

Reply: Fig. 6 has been revised and described better in the text and caption. The black crosses in Figure 6 still appear quite faint to see . . . can they be made bold and thick like they are in Figure 5?
P17 L388-389: Isn't adding to an unknown an impossibility? This needs to be explained better. Reply: Good point, we have formulated this statement differently in the revised manuscript.

— Technical comments — P2 L36: "UCL" -> "CPOM"

Reply: Done

P10 L255: ": : :**Ka and Ku** waveform: : :"

Reply: Done

P11 L264: ": : :OIB snow depth and the Warren et al. (1999) snow density **pairs**: : :"

Reply: Done

P12 L312: Define "**second-year ice** (SYI)"

Reply: Done

P15 L352: "**: : :Ka and Ku** track point: : :"

Reply: Done

Please also note the supplement to this comment:
https://tc.copernicus.org/preprints/tc-2020-196/tc-2020-196-AC1-supplement.pdf

---

## Author Comment (AC2) · 22 Nov 2020

Tonboe et al. General comments: The objective of this paper is to evaluate the capability of Ka/Ku bi-fréquency altimeters to measure the snow depth (SD) over sea-ice using a simulator. The authors tackle a particularly complex subject: what is the impact of the type of snow (salinity, density, temperature, grain size, etc.) on the performance of the measurements. The simulator is powered by measurements of terrain and its outputs are confronted with airborne measurements. This type of work is indispensable to improve the quality of

the measurements of the sea ice thickness (SIT) by satellite, and to prepare for the the Copernicus project of the CRISTAL dual-frequency altimetry satellite, one of the first missions of which is to monitor the physics and dynamics of the sea ice. As such, this work and the data used must be disseminated and made public.

Reply: Thank you for your review and pointing out where the MS could be improved. Responding to your comments has definitely improved the MS.

Nevertheless, the results presented are in contradiction with several results already published and the arguments are not sufficiently convincing.

Reply: We don't think that our simulation results are in contradiction with already published results. Actually, our simulations are confirming that we have a re-tracker sensitivity to snow (the radar penetration correction, eq. 4). We also show that there is a difference in the mean Ka- and Ku-band scattering horizons (height estimations) which is confirming the results of earlier studies. However, this Ka- and Ku-band difference is only indirectly linked with snow depth: Ka-band extinction in the snow is larger than Ku-band extinction because of the scattering from the granular structure of the snow (snow grains scattering). Deeper snow has more scattering and maybe larger snow grains, and the snow grain size is very important. Snow salinity plays a role as well but these parameters are not directly linked with snow thickness.

Indeed, although this is not explicitly stated, this study seems to conclude that the Ku-frequency almost no penetrate the snow, no matter what are the snow caracteristics (see Figure 6). In fact, most of the paper focuses on the _differential_ of snow penetration between Ka and Ku.

Reply: We don't think that we conclude anything like that. However, we understand that section 6 and figure 6 (now fig. 7) were not clear and have rewritten this section and improved the figure with attention to comments from both reviewers.

Penetrations in the snow of each individual frequency is not analyzed.

Reply: We have rewritten section 6 with attention to your point.

However, the conclusions are largely based on the measurement of the ice freeboard (FB) by means of the Ku frequency alone, a measure which appears only in this section 6 without being justified beforehand. Figure 6 in the same section is therefore difficult to interpret. For example, it is not clear how the Ku nor the green differential curve have been obtained. For SD=0 we observe an ice freeboard of 0.2m and a Ku freeboard of 0m while it does not may have a problem of penetration in the absence of snow: these 2 measures should be equal. This section 6 is far from insignificant because it leads to surprising conclusions, repeated in conclusion, including in particular the fact that the measurement of the SIT is little impacted by the method of obtaining the snow depth. This assertion is in contradiction with equation (1) of equilibrium which shows that the snow depth is involved in the process for about 30% of the measurement of the SIT (the density of snow being about 1/3 of that of water and the values of FB and SD being of the same order of magnitude). Also the model implicitly assumes that the alitimeter is in LRM mode, while all Ku altimeters currently in flight are in SAR mode. The SAR mode has a much smaller footprint than the LRM mode. It is therefore less sensitive to surface roughness and especially one cannot make the hypothesis of a retracking at 50% of the waveform (in SAR mode the retracker is between 85% to 95%). This does not call into question the study presented because the comparison of Ka/Ku penetrations is a primordial subject that deserves to be studied whatever the the altimeter mode. But it is important to mention it. And with this perspective we would like to see more precisely what are the backscatter of each of these 2 individual frequencies according to the surfaces and interfaces considered (air/snow retrodiffusion, snow/ice and volume in snow).

Reply: The different altimeter processing centers are using different re-trackers (including one of them using the $\frac{1}{2}$ power re-tracker for CryoSat-2 data processing) and all processing centers are correcting for radar penetration in the snow meaning that the re-tracker is sensitive to snow. Otherwise they would not have to do that. Our model

simulation experiments are confirming that the track-point is sensitive to snow in line with the practice of the different processing centers.

Finally, this multilayer model seems to consider only one layer for snow, whereas we generally consider at least 2 layers for snow over sea ice, with a hard and dense superciel layer and a deepest layer of very metamorphosed grains of consequent dimensions (of the order of centimeters). This point should also be discussed. I would therefore recommend to the authors to deepen the presentation of the measures carried out, and especially the model deployed and the conclusions that it brings on each of the frequencies, quite to reduce the part 6 on the results expected by altimetry.

Reply: We have detailed the discussion of justifying the one layer set-up vs. the multilayer set-up in the 5 detailed profiles in the text. We agree that the one layer set-up is a simplification compared to reality. However, it allows us to study the direct effect of snow depth on the track-point and on the Ka- and Ku-band track point difference. This would not be possible with a multilayer set-up and actually the simulations with the 5 multilayer profiles are largely confirming the simulations with the one-layer set-up.

Detailed comments: P1 L27: I do not agree with the following sentence: "... the impact of using a snow climatology versus the actual snow depth is relatively small on the measured freeboard" that must be more clearly demonstrated (see general comments and other comments bellow).

Reply: In the processing of radar altimeter data for deriving the radar freeboard (often assumed coincident with the snow ice interface) there are two corrections involving snow: 1) the buoyancy correction, eq. 1, 2) and the radar snow propagation correction, eq. 4. That's it. These two corrections almost cancel out and that is why the impact of snow is small in the processing.

P2 L45: "The radar scattering horizon or track point is conceptualized as the scattering surface depth detected by the radar re- tracker algorithm and the floe buoyancy" : this study should not depend on the buoyancy but only on the penetration.

Reply: We have to deal with both the radar penetration and the buoyancy at the same time because these two effects largely cancel out (fig. 7).

P2 L51: The following sentence is true only for the heuristic retrackers, not for the retrackers based on physical models: "The re-tracker algorithm can be tuned so that the radar scattering horizon coincides with the snow/sea ice interface."

Reply: Ok, we added that tuning is possible with the re-tracker that we are using. Bias correction is always possible with any re-tracker even though this is not removing the track point sensitivity to snow. All processing centers (using different re-trackers) are doing the same radar penetration correction (eq. 4) which means that the different re-trackers are all sensitive to snow. This sensitivity can be illustrated with a simple re-tracker and this is what we do here.

P2 L54: What do you mean by : "leading to preferential sampling of the thinner ice types " ?

Reply: We have rephrased this statement in the text because both reviewers had comments about the term "preferential sampling". Radar backscatter from thin ice is orders of magnitude larger, in areal fraction, than backscatter from thick ice and when both are present within the footprint. As such, the waveform is dominated by the thin ice backscatter disproportional to its areal fraction. This is preferential sampling. It is described in Tonboe et al. 2010 (reference list). Thin ice backscatter can be detected because of its specular backscatter but the disproportional sampling of the radar also happens with ice surface types within the footprint which are not easily detected. P2 L61: when speaking of "penetration correction" do you include the speed propagation reduction into the snow ? Reply: Yes, the speed of light in the snow is a function of its permittivity. It is included in the model. This is mentioned in connection with the description of equation 2.

P6 L142: You say that the "surface roughness is assumed to not influence the scattering horizon variability in our model simulations" while the surface as a strong impact on

the altimetric waveforms. Does that mean that the model do not reflect the altimetric behavior? Please comment.

Reply: The model does include surface roughness as a parameter (F in Tab. 1) but it is constrained to one value in the simulations and therefore it does not influence the scattering horizon variability. We do not assume that roughness does not influence the scattering horizon variability, because it does, but not in these simulation experiments. That is also stated in the text.

P7 Fig 3: Please specify that depth=0.0 corresponds to the bottom, not to the surface! (if I dont mistake)

Reply: Thanks, good point, it has been included in the text.

P9 L206: You say that "The track point is found at half of the maximum waveform power point in time". It is a true mean for LRM altimetry but physical retrackers show that this value varies according to the roughness and the specularity of the surface. For SAR altimetry the mean value is much higher.

Reply: The justification of using this re-tracker threshold is given in the sentence that follows: " While different track point thresholds will shift the scattering horizon vertically (Ricker et al., 2014), the location of the scattering horizon does not change the modeled sensitivity to snow depth (Tonboe, 2017)." Using our model, we do not see that the different levels of the track-point changes the sensitivity to snow (Tonboe, 2017) and the roughness and "specularity" is constant in our simulations.

P9 L210: What do you mean by "the total backscatter is dominated by surface/interface scattering"? The interface is between the surfaces? Or it is another surface? Do you mean that the volum scattering is negligeable? In such a case it must be said/shown explicitely.

Reply: In the model we deal with two types of scattering: 1) surface scattering from the plane interfaces (air-snow, snow-ice and snow layering), and 2) volume scattering

from the granular structure or particles in the snow. Yes, surface/interface scattering dominates and volume scattering is negligible as a backscatter source. However, vol. scattering contributes to extinction in the snow and therefore to the magnitude of the snow ice interface scattering. Volume scattering is described in lines 230-233 of the original version of the manuscript.

P9 L213: In the sentence "This assumption is believed to be more realistic than other sea ice surface scattering" please specify which other sea ice surface scattering you are thinking off.

Reply: Thanks, the geometric optics model is building on different assumptions than the flat-patch model that we are using, yet the two models have quite similar predictions (Fetterer et al. 1992). We have specified that in the revised version.

P10 L241: "the track point is computed as a point in time located midway between the noise floor and the maximum return signal power received by the radar." This is pertinent only for LRM altimetry.

Reply: This re-tracker is used for SAR altimeters as well.

P11 Table1: Only one layer for the snow. Is it realistic? Please comment.

Reply: Snow on sea ice can be layered and therefore we included the 5 snow profiles from the Canadian Arctic in addition to the one layer set-up in Tab. 1. In fig 7 you can see that the simulations of the layered profiles line up with the one-layer set-up especially for snow depths less than 20 cm. To answer your question: yes, it is realistic, the one layer set-up does give similar simulation results as seen in fig 7. This is also commented in the text discussing fig. 7.

P12 L297: Typo: "Ku- and Ku-"

Reply: Thanks.

P14 L338: "The snow climatology is used to 1) compensate for the effect of the snow

cover on the ice floe buoyancy, and 2) to compute radar propagation in the snow." For point 2) I suppose that you mean "compute radar slow down speed propagation in the snow" ?

Reply: Thanks, yes, that is what is meant. It has been clarified in the text.

P15 L345: "We do not show the actual ice thickness, but it is proportional to the freeboard." As shown by your equation (1) the SIT depends also on the snow load. So please could you precise the SIT used in the Fig 6.

Reply: Yes, the thickness is written in the figure text as well. The ice is 2m thick.

P15 L352: "The green line in Figure 6 shows the combined effect of snow on the track point and the floe buoyancy." Which track point? Ka? Ku? Until this section 6 only the ka-ku difference has been considered.

Reply: It's the Ku-band track point. This is written in the figure text, and it has now been included it in the legend in the revised text.

P15 L357: "The effect of snow depth on the Ku- and Ka- track point is linear up to snow depths of âĹij 50 cm (Figure 6)." Fig 6 does not show the Ka measurement.

Reply: Thank you. We have now added both the Ka- and Ku-band track-points in figure 7 and described that in the text.

P15 L360: "The correction" for the track point is on average 0.35 times the snow depth". Which track point? Ka? Ku?

Reply: Thanks, it's Ku. . . it has now been specified.

P15 L368: typo "SYI"

Reply: It was not, but should have been defined earlier in the text at L312. SYI is second year ice... It has been defined in the text.

P16 Fig 6: "Red circles is the Ku-band radar track point as a function of snow depth

and density" : The Ku FB has not been introduced beforhand, how do you obtain it? "The combined effect of both Ka- and Ku-band track point and buoyancy is the green line freeboard": how it is computed ? How do you get a nul Ku-FB for a ice-FB, without snow, of 20cm? Thus it is clearly not a problem of snow penetration!

Reply: The freeboard is 21cm when there is no snow (blue line). We realize that this is a busy figure and we have tried to make it clearer.

P17 L389: "the snow climatology results in a small impact on the derived sea ice thickness": this sentence is clearly in contradiction with equation (1). See general comments.

Reply: Equation 1 is describing the floe buoyancy and it is not including the correction for radar penetration in the snow. Adding the two together gives a small impact on the effect of snow.

P17 L393: "The small impact of the snow on the measured freeboard is the reason why the sea ice thickness can be derived using radar altimeters even without actual snow information." If the first part of this sentence could be true, the second one is clearly false. Even if we can not measure precisely the FB, the SD does have nevertheless a strong impact on the resulting SIT! It is easy to demonstrate using equation (1) and various SD datasets.

Reply: There is equation 1 and the radar penetration correction. Both are included in radar altimeter data processing and they largely cancel out. This is merely confirmed by our model simulations.

P17 L405: "Our simulations demonstrate that the direct Ka- and Ku-band track point difference sensitivity is about 0.033 times the snow": it is not (yet) a demonstration but still an assumption based on a model. Please mitigate.

Reply: We have changed the wording here from “demonstrate” to ”shown” It has been shown that there is a Ka- and Ku-band radar freeboard difference on SIT estimation

from radar altimetry and that this can be linked with snow depth: e.g. Armitage and Ridout, 2015, Guerreiro et al., 2016, Lawrence et al., 2018 What we show is that this difference is not solely a function of snow depth, but rather both scattering and absorption processes leading to enhanced extinction in the snow. Snow salinity, snow grain size inhomogeniety are further linked with different ice types.

P18 L421: "This implies that the measured freeboard is nearly independent of snow depth." Using Ka? Ku? Both? Please be more precise or mitigate.

Reply: Yes, that is the point. We reformulated the sentence to explain better.

P18 L424: "the impact of actual snow depth is small in the sea ice thickness estimate": equ (1) shows that the SD may not be negligeable at all.

Reply: Again, considering both equation 1 and the radar penetration correction, these largely cancel out.

Please also note the supplement to this comment:
https://tc.copernicus.org/preprints/tc-2020-196/tc-2020-196-AC2-supplement.pdf

---

## Author Response (AR1)

Dear Dr. Tonboe, dear co-authors,

In my view, the reviewers have offered excellent comments and made valid points following a very thorough read. At this stage, please, submit your revised manuscript. I hope you will make your edits in track-changes, because in your responses to the reviewers it was not always clear to me what improvements would eventually be made in the manuscript.

I note that both reviewers had the impression that your results were contradicting the literature. I hope appropriate clarifications will be made in the manuscript to ensure our readership will not have the same concern when reading your study. A single sentence in the conclusion seems insufficient.

Regards

Dear Ludovic Brucker, dear reviewers, we sincerely appreciate the reviewers comments and we have tried to address all comments and improve the MS where it was not clear. All our changes are marked with track changes in the MS. We are not contradicting results from earlier studies about the Ka- and Ku-band track-point difference, we are just specifying the mechanisms behind this difference. Also that has been clarified in the revised MS.

Best regards Rasmus

---

## Author Response (AR2)

**Reply to 3[rd] anonymous reviewer**

We would thank the reviewer for raising the question about the reference height in Figure 7. We agree with the reviewer that the track point should be within the snow cover. We have changed the reference height to be more in line with the way ice thickness is derived from satellite altimetry and now the track point is within the snow cover (see argumentation below).

*The authors addressed the suggestions and comments by the reviewers in a adequate way and I feel this improved the manuscript a lot compared to its earlier version. The authors did a good job in providing answers to the open questions and also did some additional changes to the manuscript as visible from the ATC document that improve the overall readability (such as using SI units like m instead of cm) and clarifying Figures. Aside from a question/remark and a technical thing I recommend publication.*

*Comments:*

*A remark to Reviewer 1's question/remark on Figure 5 (now 6) and the attached new version by the authors showing the penetration depth:*

*I found it a bit surprising that the simulated trackpoints using your 50% retracker threshold are more or less always(?) already inside the sea ice (i.e. below the snow/ice interface) instead of what we would commonly assume, i.e. inside the snow pack.*

Reply: Thanks for pointing this out. The reason for the track point to be located "inside the sea ice" is that we are using the real water surface height as a reference and not the re-tracked water surface height. We have used the model and the re-tracker to simulate the lead surface height (re-tracked water surface). The difference between the real and the re-tracked lead surface height is 0.187 m. We agree that using the real water surface height as a reference is inconsistent with the way ice thickness is derived using radar altimeters and we have therefore added the offset to the re-tracked ice surfaces in Figure 7, with associated text description. It does not change the conclusions, but only an upward shift in the re-tracked surface height by 0.187 m.

*Also the difference between Ka/Ku track points is almost for all measurements very small or non-existent which reduces hopes for CRISTAL kind of. Could the authors comment on this?*

Reply: The snow depth in itself does not have a large impact on the Ka- and Ku-band track point difference and other variables which are to some extent related to snow depth are playing a role as well in creating the observed difference (the observations reported in the literature). In order to use the Ka- and Ku-band track point difference for deriving the snow depth, we think that we need to understand the underlying processes in more detail. Lines 316-324 are describing this.

*Does this impact or result from the overall model performance? In other comparison studies, AWI measurements (who use this kind of retracker threshold in production) tend to overestimate freeboard*

*rather than underestimate it compared to other producers (who use a higher threshold in the 80-95% range).*

Reply: The reason for this offset was the difference between the real and the retracked water surface height reference (see comment above).

*Furthermore, I found there to be several instances of inconsistencies in capitalization,e.g. between Figure 7 and figure 7 on P7 L 360 and several other occurrences. I feel this is work that could be changed by the authors and not solely by the copy-editing.*

Reply: Thanks for pointing this out. In addition to L360 we found two cases where a reference to an equation was not capitalized and a couple of minor typos (see MS with track changes).